

# Prognostic significance of thyroid hormone T3 in patients with septic shock: a retrospective cohort study

Caizhi Sun[*], Lei Bao[*], Lei Guo, Jingjing Wei, Yang Song, Hua Shen and Haidong Qin

Department of Emergency, Nanjing First Hospital, Nanjing Medical University, Nanjing, Jiangsu Province, China
[*] These authors contributed equally to this work.

Corresponding author
Haidong Qin, icuqhd@163.com

## ABSTRACT

**Background.** The role of thyroid hormones is crucial in the response to stress and critical illness, which has been reported to be closely associated with a poor prognosis in patients admitted to the intensive care unit (ICU). This study aimed to explore the relationship between thyroid hormone and prognosis in septic shock patients.

**Methods.** A total of 186 patients with septic shock were enrolled in the analytical study between December 2014 and September 2022. The baseline variables and thyroid hormone were collected. The patients were divided into survivor group and non-survivor group according to whether they died during the ICU hospitalization. Among 186 patients with septic shock, 123 (66.13%) were in the survivor group and 63 (33.87%) were in the non-survivor group.

**Results.** There were significant differences in the indictors of free triiodothyronine (FT3) ($p = 0.000$), triiodothyronine (T3) ($p = 0.000$), T3/FT3 ($p = 0.000$), acute physiology and chronic health evaluation II score (APACHE II) ($p = 0.000$), sequential organ failure assessment score (SOFA) ($p = 0.000$), pulse rate ($p = 0.020$), creatinine ($p = 0.008$), PaO2/FiO2 ($p = 0.000$), length of stay ($p = 0.000$) and hospitalization expenses ($p = 0.000$) in ICU between the two groups. FT3 [odds ratio (OR): 1.062, 95% confidence interval(CI): (0.021, 0.447), $p = 0.003$], T3 (OR: 0.291, 95% CI: 0.172-0.975, $p = 0.037$) and T3/FT3 (OR: 0.985, 95% CI:0.974-0.996, $p = 0.006$) were independent risk factors of the short-term prognosis of septic shock patients after adjustment. The areas under the receiver operating characteristic curves for T3 was associated with ICU mortality (AUC = 0.796, $p < 0.05$) and was higher than that for FT3 (AUC = 0.670, $p < 0.05$) and T3/FT3 (AUC = 0.712, $p < 0.05$). A Kaplan-Meier curve showed that patients with T3 greater than 0.48 nmol/L had a significantly higher survival rate than the patients with T3 less than 0.48 nmol/L.

**Conclusions.** The decrease in serum level of T3 in patients with septic shock is associated with ICU mortality. Early detection of serum T3 level could help clinicians to identify septic shock patients at high risk of clinical deterioration.

## INTRODUCTION

Sepsis refers to a life-threatening syndrome caused by a dysregulated host response to infection that may progress to fatal shock and contributes to 11.0 million deaths worldwide in 2017, representing the mortality closing to 20% worldwide (*Rhodes et al., 2017*; *Singer et al., 2016*; *Rudd et al., 2020*). Alarmingly, the prevalence and mortality rate is still continuing its upward trend (*Suarez De La Rica, Gilsanz & Maseda, 2016*). Septic patients frequently require expensive and intensive treatments, which estimated to be $27,461 per case, thus making sepsis a major public health concern (*Fleischmann et al., 2016*; *Arefian et al., 2017*; *Dietz et al., 2017*). Sepsis is associated with dysfunction of multiple organs and hypotension or shock tending to contribute to hypoxia, which would reduce the ability of cells to produce ATP. Thyroid hormones play a crucial role in the response to stress and critical illness. Therefore, the impact of severe systemic diseases on thyroid metabolism is increasingly emphasised by researchers. Decreased thyroid hormones level has often been called low T3 syndrome or euthyroid sick syndrome or nonthyroidal illness syndrome (NTIS) (*Liu et al., 2016*; *Van den Berghe, 2014*), which has been described as the key role of thyroid hormones in counteracting biochemical catabolism leading to sepsis or septic shock (*Warner & Beckett, 2010*).

The relationship between thyroid hormone and the severity of sepsis/septic shock is getting more and more attention (*Langouche, Jacobs & Van den Berghe, 2019*; *Dietrich et al., 2008*; *Bertoli et al., 2017*). A study from China showed that free riiodothyronine (FT3) could be used as a predictor of all-cause mortality in ICU patients more than a decade ago, and the predictive efficacy was further improved when combined with the APACHE II score (*Wang et al., 2012*). However, the study did not specifically observe the effect of thyroid hormones on outcomes in patients with septic shock. Subsequent research found that early detection of serum FT3 and FT4 levels could help clinicians to identify patients at high risk of clinical deterioration (*Liu et al., 2021*). Nevertheless, the study enrolled smaller numbers of critically ill patients and the origin of septic shock patients was limited to EICU, which could not be representative of most ICU septic shock patients in China and around the world. Considering the aforementioned condition, whether current thyroid hormone level could be used as a prognostic indicator of septic shock patients is still controversial and needed further study.

Therefore, further research is needed to clarify the association between NTIS and the prognosis of septic shock patients in the ICU. The aim of this study was to undertake a investigation of septic shock patients admitted to all ICU in our hospital to identify the prognostic value of thyroid hormones levels and provide reference for clinical research of thyroid hormone in the treatment of septic shock.

## PATIENTS AND METHODS

### Participants

This single-center, retrospective study recruited patients admitted to the ICU of Nanjing Hospital Affiliated to Nanjing Medical University between December 2014 and September 2022. The septic shock was defined according to the diagnostic criteria of Sepsis and

Septic Shock 3.0 published in 2016 (*Rhodes et al., 2017*). The exclusion criteria were as follows: (1) Patients with less than 24 h stay or more than 30 days in ICU, (2) patients with hypothalamic-pituitary disease, (3) patients administered with antithyroid drugs or other iodine containing drugs in one month, *e.g.*, miodarone and glucocorticoid, (4) patients with severe immunodefciency and hypothyroid or hyperthyroid diagnosis history, (5) age <18 years. The enrolled patients were divided into survivor group and non-survivor group according to whether they died during the ICU hospitalization. The survivor group was defined as patients who improved after treatment and were discharged or transferred from ICU to the general ward. The non-survivor group was defined as patients who died after treatment during the ICU hospitalization.

## Data collection

*Datebase*—Data for the study analysis were derived from the electronic medical records (EMR) which include demographic information, comorbid diseases such as coronary heart disease (60.22%, 112/186), hypertension (36.56%, 68/186), diabetes mellitus (26.88%, 50/186) and cerebrovascular disease (17.74%, 33/186). The infected site of septic shock leading to ICU admission such as respiratory infection (4.09%, 82/186), urinary tract infection (23.12%, 43/186), bloodstream infection (7.53%, 14/186), abdominal cavity infection (7.53%, 14/186) and others (4.30%, 8/186) were recorded.

Blood samples were taken from the patients on the first day of the ICU admission for detection of laboratory indicators and thyroid hormone, before the drug treatment started. Levels of FT3, FT4, T3, T4 and TSH were measured on the next morning after ICU admission by the chemiluminescent immunometric assay method using an Abbott ARCHITECT i2000 analyzer (Abbott, Chicago, IL, USA) in the Department of Nuclear Medicine of our hospital. In order to ensure the consistency of T3 and FT3 units, T3*1000/FT3 is taken as the ratio of T3/FT3. According to the manufacturer's instructions, the normal reference intervals used were: 2.63−5.7 pmol/L for FT3, 0.89−2.44 nmol/L for T3, 9.0−19.0 pmol/L for FT4, 62.68–150.84 nmol/L for T4 and 0.35−4.94 mIU/L for TSH.

The septic shock patients' vital signs and related laboratory indicators were recorded upon admission to the ICU, including systolic blood pressure (SBP), diastolic blood pressure (DBP), mean arterial pressure (MAP), blood gas analysis, white blood cell (WBC), procalcitonin (PCT), interleukin-6 (IL-6), platelet (PLT) and creatinine. Blood gas analysis including the indicator of lactic acid were detected in the ICU using a multifunction blood gas analyzer (Rayto, USA). The other laboratory INDICATORS were detected by the Laboratory Department of Nanjing First Hospital.

The acute physiology and chronic health evaluation II score (APACHE II) and sequential organ failure assessment score (SOFA) were completed within 24 h of admission to the ICU for the enrolled patients. The duration of mechanical ventilation, length of stay in ICU, whether to use continuous renal replacement therapy (CRRT) and costs of hospitalization were recorded.

## Research ethics

This study protocol was reviewed and approved by the Institutional Review Board of Nanjing Hospital Affiliated to Nanjing Medical University (No. KY20201102-03) and was

conducted in accordance with the Declaration of Helsinki. The requirement of informed consent for the study was exempt due to restrained database access for analysis purposes only.

## Statistical analyses

All data were analyzed using SPSS 20.0. Data were presented as median and interquartile ratio (IQR) or numbers, as appropriate. Continuous and categorical variables were compared using Mann–Whitney U test. Univariate and multivariable logistic regression analysis were conducted to identify the independent risk factors for ICU mortality. The receiver operating characteristic (ROC) curves for the ability of thyroid hormone to predict the ICU mortality were analyzed. Kaplan–Meier analysis was used to estimate the probability of survival and comparison was made by Log-rank test. A $P$ value of <0.05 was considered statistically significant.

# RESULTS

## Baseline of patient characteristics

There were two hundred and forty-seven patients who met the septic shock definition by Sepsis-3. Sixty-one patients were excluded based on exclusion criteria, resulting in a cohort of one hundred and eighty-six patients, including one hundred and twenty-three patients in the survivor group and sixty-three patients in the non-survivor group (Fig. 1). The median age in survivor group was 72.38 ± 9.64 years with 61.79% being men and the non-survivor group was 74.57 ± 8.57 years with 69.84% being men. The baseline and clinical characteristics of the study population upon shock recognition were collected, including demographic information, infection sites leading to ICU admission, past medical history, vital signs, APACHE II score at shock recognition and laboratory indicators. The non-survivor group had a higher APACHE II score, SOFA score and creatinine level (30.28 ± 6.67 $vs$ 24.03 ± 5.90, $p = 0.000$; 12.14 ± 3.00 $vs$ 8.67 ± 3.12, $p = 0.000$; 299.38 ± 125.19 $vs$ 248.20 ± 123.59, $p = 0.008$), while the PaO2/FiO2 in the non-survivor group was significantly lower than that in the survivor group [165.43 (98.63, 200.39) $vs$ 217.54 (198.63, 267.32)]. In addition, no significant difference was found in infection sites, past medical history, WBC, PCT, IL-6 and albumin and so on. Details of the two groups are shown in Table 1.

## Serum thyroid hormone levels

Blood samples were taken from 186 patients with septic shock on the first day of the ICU admission for thyroid hormone detection. No statistical difference of the serum TSH, FT4 and T4 seen between non-survivor and survivor groups [0.80 ± 0.79 $vs$ 0.64 ± 0.61, $p = 0.175$; 12.19 (9.78,12.84) $vs$ 12.32 (10.22,12.49), $p = 0.575$; 69.41 (63.02,75.30) $vs$ 74.93 (55.75,83.03), $p = 0.209$]. While the levels of FT3, T3 and T3/FT3 in non-survivor group were significantly lower than that in survivor group (all $p = 0.000$; Fig. 2).

## Comparison of outcomes

The mechanical ventilation (MV) rate was higher in non-survivor group than that in the survivors group ($\chi^2 = 12.148$, $p = 0.000$). The length of ICU stay in the survivor group

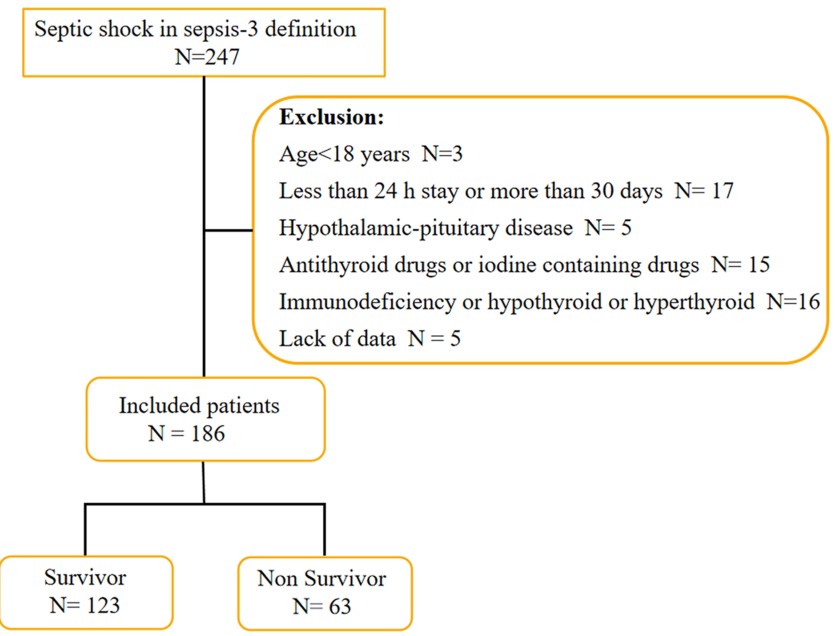

**Figure 1** Flow chart of patients in the study cohort.

was longer and renal replacement therapy rate decreased significantly than that in the non-survivor group [7.00 (5.00,11.00) *vs* 5.00 (2.00, 9.00), $p = 0.000$; 6.50% *vs* 36.51%, $\chi^2 = 11.600$, $p = 0.000$]. It is worth noting that the costs of hospitalization in ICU were significantly higher in the non-survivor group than that in the survivor group (6.14 ± 4.67 *vs* 8.73 ± 8.23, $p = 0.000$). Details of the two groups are shown in Table 2.

## Factors associated with ICU mortality

The multivariable logistical regression analysis suggested that FT3, T3, T3/FT3, Creatinine and PaO2/FiO2 were independent factors for ICU mortality (FT3: OR = 0.097, 95%CI [0.021–0.447], $p = 0.075$; T3:OR =0.291, 95%CI [0.172–0.975], $p = 0.037$; T3/FT3: OR = 0.985, 95%CI [0.974–0.996], $p = 0.006$; Creatinine: OR =1.004, 95%CI [1.001–1.008], $p = 0.019$; PaO2/FiO2: OR = 0.986, 95%CI [0.979–0.993], $p = 0.000$), as shown in Table 3.

## Predictive value of thyroid hormones for ICU mortality

To investigate whether thyroid hormone is useful in predicting ICU mortality in septic shock patients, ROC analysis was conducted using clinical and laboratory indicators. Among the SOFA, serum FT3, T3 and T3/FT3, AUC for serum T3 (AUC 0.796,95% CI [0.727–0.864], $p = 0.000$) was higher than the AUC for FT3 (AUC 0.670, 95% CI [0.587–0.753], $p = 0.000$) and T3/FT3 (AUC 0.712, 95% CI [0.635–0.788], $p = 0.000$) and SOFA (AUC 0.779, 95% CI [0.712–0.845], $p = 0.000$), as shown in Fig. 3. With a cut-off value of 0.48 nmol/L of T3 determined on the ROC curve in the derivation cohort and Kaplan -Meier survival curves were established (Fig. 4). The septic shock patients' survival rates were significantly different when stratified according to serum T3 level on day of ICU

**Table 1  Baseline and clinical characteristics of the study population.**

| Patients characteristics | Survivors ($n = 123$) | Non-survivors ($n = 63$) | P |
|---|---|---|---|
| Male, n (%) | 76 | 44 | 0.277 |
| Age, mean ± SD | 72.38 ± 9.64 | 74.57 ± 8.57 | 0.199 |
| Infection sites leading to ICU admission, n(%) | | | 0.761 |
| Respiratory infection | 55 (44.72%) | 27 (42.85%) | |
| Urinary tract infection | 27 (21.95%) | 16 (25.40%) | |
| Bloodstream infections | 11 (8.94%) | 3 (4.76%) | |
| Abdominal cavity infection | 24 (19.51%) | 15 (23.81%) | |
| Others | 6 (4.88%) | 2 (3.17%) | |
| Past medical history, n (%) | | | 0.877 |
| Coronary heart disease | 65 (85.53%) | 47 (74.60%) | |
| Hypertension | 42 (55.26%) | 26 (41.27%) | |
| Diabetes mellitus | 32 (42.11%) | 18 (28.57%) | |
| Cerebrovascular disease | 19 (25%) | 14 (22.22%) | |
| APACHE II score, mean ± SD | 24.03 ± 5.90 | 30.28 ± 6.67 | 0.000 |
| SOFA score, mean ± SD | 8.67 ± 3.12 | 12.14 ± 3.00 | 0.000 |
| Vital signs on ICU admission | | | |
| SBP, mmHg, median (IQR) | 82.00 (78.00, 87.00) | 85.00 (79.00, 90.00) | 0.146 |
| DBP, mmHg, median (IQR) | 50.00 (48.00, 60.00) | 50.00 (45.00, 60.00) | 0.273 |
| MAP, mmHg, median (IQR) | 61.67 (57.33. 68.33) | 61.67 (56.67, 70.00) | 0.915 |
| Pulse rate, beats/min,, median (IQR) | 123.00 (104.00, 139.00) | 147.00 (105.00, 153.00) | 0.020 |
| Body temperature, °C, mean ± SD | 38.42 ± 0.71 | 38.59 ± 0.71 | 0.137 |
| Respiratory rate, breaths/min, mean ± SD | 33.87 ± 9.77 | 35.02 ± 11.43 | 0.476 |
| Laboratory indicators | | | |
| WBC, ×$10^9$/L, mean ± SD | 16.32 ± 8.18 | 14.40 ± 7.17 | 0.116 |
| PCT, ng/mL, median (IQR) | 16.27 (5.31, 35.10) | 16.87 (5.52, 35.29) | 0.368 |
| IL-6, pg/mL, median (IQR) | 84.27 (27.38, 243.00) | 129.91 (26.31, 1015,24) | 0.086 |
| Platelet, ×$10^9$ /L, median (IQR) | 90.00 (65.00, 190.00) | 140.00 (57.00, 210.00) | 0.390 |
| Albumin, g/L, median (IQR) | 30.00 (27.00, 43.00) | 31.00 (27.00, 40.00) | 0.539 |
| Creatinine, $\mu$mol/L, mean ± SD | 248.20 ± 123.59 | 299.38 ± 125.19 | 0.008 |
| PaO2/FiO2, mmHg, median (IQR) | 217.54 (198.63, 267.32) | 165.43 (98.63, 200.39) | 0.000 |
| Lactate, mmol/L, median (IQR) | 11.40 (8.80, 12.60) | 11.40 (9.40, 12.80) | 0.653 |

Notes.

APACHE II score, acute physiology and chronic health evaluation II score; ICU, intensive care unit; WBC, White blood cell; PCT, Procalcitonin; SBP, Systolic blood pressure; DBP, Diastolic blood pressure; MAP, Mean arterial pressure; SOFA score, Sequential Organ Failure Assessment score; IL-6, interleukin-6.

admission and the patients with higher serum T3 level had better survival than that with lower serum T3 level.

## DISCUSSION

In this retrospective study, we sought the potential relationship between thyroid hormone levels measured in the first 24 h after ICU admission and prognosis in patients with septic shock admitted to the intensive care unit (ICU) . The results showed that the mortality of septic shock patients admitted to the ICU is 33.87% (63/186), which is consistent with

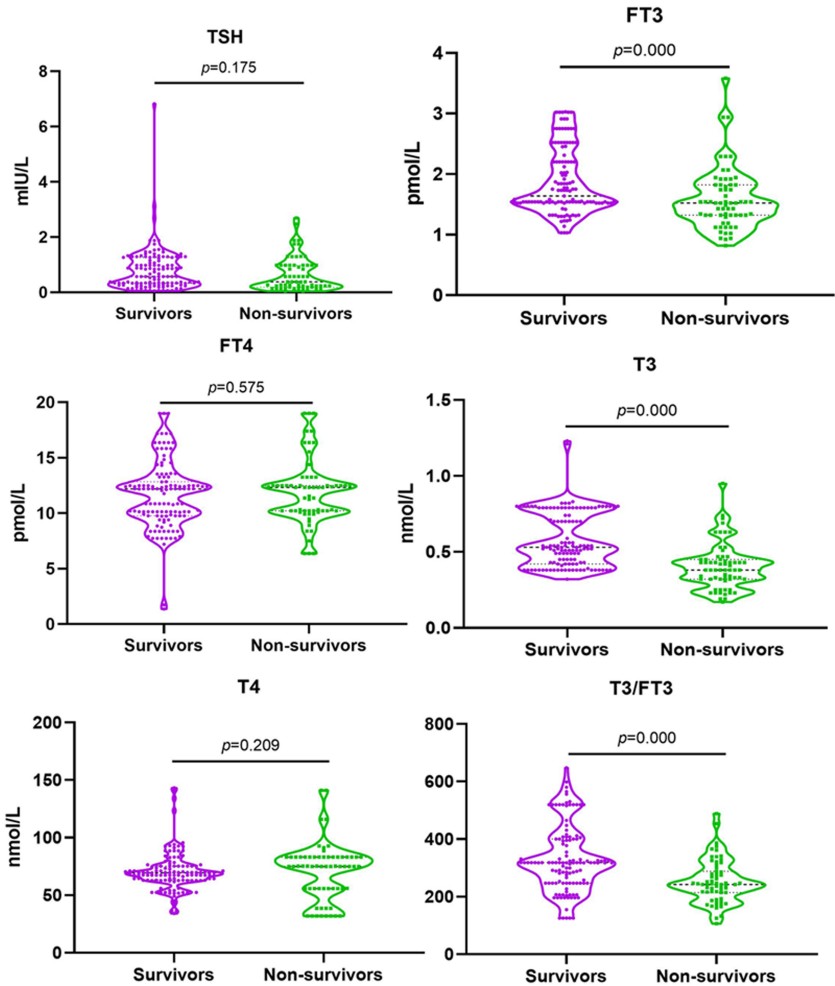

**Figure 2** **Serum level of FT3, FT4, TSH, T3,T4 and T3/FT3 in survivors and non-survivors.** Abbreviations: FT3, free triiodothyronine; FT4, free thyroxine; T3, triiodothyronine; T4, thyroxine; TSH, thyroid stimulating hormone. Comparison between the survivors and non-survivors groups using independent sample T test. $P < 0.05$.

**Table 2 Outcomes of the two groups.**

| Outcomes | Survivors ($n = 123$) | Non-survivors ($n = 63$) | P |
|---|---|---|---|
| MV rate,n (%) | 61 (49.59%) | 48 (76.19%) | 0.000 |
| Duration of MV (d) | 5.28 ± 3.77 | 5.94 ± 4.88 | 0.303 |
| Renal replacement therapy, n (%) | 18 (14.63%) | 23 (36.51%) | 0.001 |
| Length of ICU stay (d), median (IQR) | 7.00 (5.00,11.00) | 5.00 (2.00, 9.00) | 0.000 |
| Hospitalization expenses in ICU, mean ± SD, (×10,000 yuan) | 6.14 ± 4.67 | 8.73 ± 8.23 | 0.000 |

**Notes.**

MV, mechanical ventilation; ICU, intensive care unit; IQR, interquartile range.
**Table 3  The performance of FT3, T3 and T3/FT3 at admission for predicting ICU mortality in ICU septic shock patients.**

| Indicators | $\beta$ | $S_{\bar{x}}$ (S.E.) | Wald | P | OR | 95% CI |
|---|---|---|---|---|---|---|
| APACHE II | 0.06 | 0.034 | 3.159 | 0.075 | 1.062 | (0.994, 1.135) |
| SOFA | 0.349 | 0.061 | 33.128 | 0.000 | 1.418 | (1.259, 1.597) |
| FT3 | −2.337 | 0.782 | 8.929 | 0.003 | 0.097 | (0.021, 0.447) |
| T3 | −0.534 | 2.986 | 0.006 | 0.037 | 0.291 | (0.172, 0.975) |
| T3/FT3 | 0.015 | 0.006 | 7.611 | 0.006 | 0.985 | (0.974, 0.996) |
| Creatinine | 0.004 | 0.002 | 5.468 | 0.019 | 1.004 | (1.001, 1.008) |

**Notes.**

Adjusted for age, gender, baseline APACHE II score and SOFA score.

APACHE II score, acute physiology and chronic health evaluation II score; SOFA score, sequential organ failure assessment score; OR, odds ratio; CI, confidence interval.

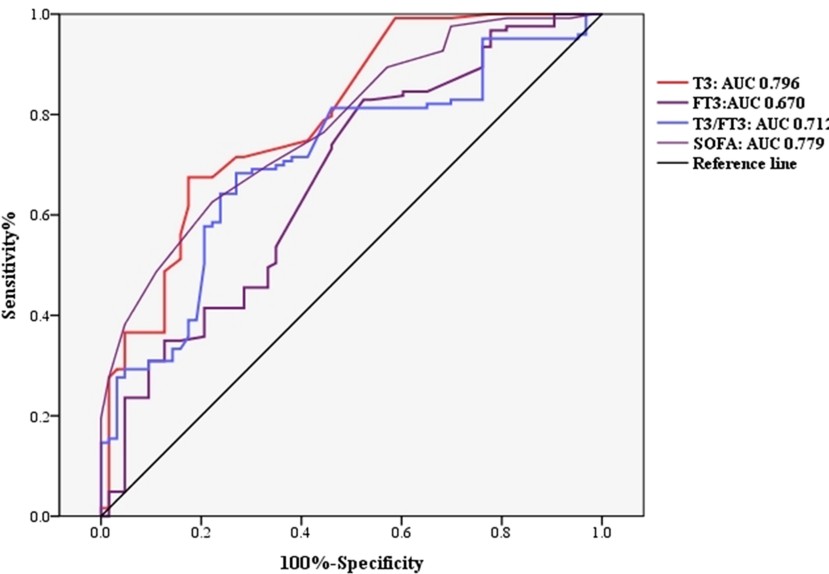

**Figure 3  Receiver operating characteristic (ROC) curve analysis.** Abbreviations: FT3, Free triiodothyronine; T3, Triiodothyronine; SOFA score, Sequential organ failure assessment score. ROC curves for ICU mortality prediction of adult patients with septic shock by the serum FT3, T3 and T3/FT3 at admission. The area under the ROC curve (AUC) is shown.

the previous study (*Chambers et al., 2018*). In addition, there was a significant lower level of both T3 and FT3 in the non-survivors than that in the survivors. However, we did not find any significant differences in terms of serum TSH, FT4 and T4 levels at admission between the two groups. Although the topic in this study is similiar to the existed studies, some of the differences observed in the present study may be attributed to more patients and statistical indicators included and the difference in the term definition of "survivor and non survivor" (*Liu et al., 2021* and *Cornu et al., 2020*). What is noteworthy is that the length of ICU stay were shorter in the non-survivors, although the ICU hospitalization expenses were higher, which may be due to non-survivors requiring longer periods of CRRT and mechanical ventilation, higher doses of vasoactive drugs besides more critical illness and poorer treatment response.

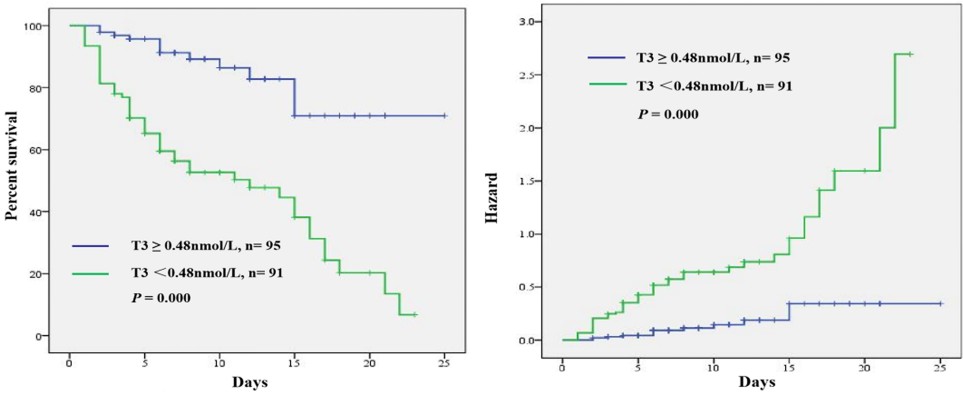

**Figure 4  Kaplan Meier survival curves of 186 adult patients with septic shock based on the T3 cut-off value 0.48 nmol/L on day of ICU admission.** A significant difference was measured between the two curves ($p < 0.01$, log-rank test).

The univariable analyses suggested that the mortality of septic shock patients admitted to ICU was closely associated with APACHE II score, SOFA score, PaO2/FiO2, FT3, T3 and T3/FT3 levels. The ROC analysis showed that the thyroid hormone T3 has a higher specificity of 82.6% as a single indicator, which is of great prognostic value compared to FT3, T3/FT3 and SOFA score. Higher T3 specificity tends to indicate low false-positive rates. In clinical practice, T3 is a single laboratory test and might provide information about the prognosis of septic shock patients quickly and objectively, which of the predictive value is higher than the indicators of APACHE II, SOFA, FT3 and T3/FT3. Moreover, a cut-off value of 0.48nmol/L T3 indicated a higher risk of ICU mortality.

As we all know, APACHE II score is based on the worst data obtained within the first post-admission and is not recalculated during the patients' stay. Therefore, the higher the scores reached, the higher the risk of critically ill patients in-hospital mortality. APACHE II score system is considered as 'gold standards' in prognostication among severely ill patients in individual ICUs worldwide, due to including lager numbers of indicators that offer significant advantages over a single indicator, and no single indicator has been reported to date to exceed the APACHE II score in terms of the prognostic value for mortality (*Vincent & Moreno, 2010*; *Haq et al., 2014*; *Alizadeh et al., 2014*; *Lee et al., 2017*; *Ryan et al., 2016*). In the present study, we found that the APACHE II score in the non-survivors group was significantly higher than that in the survival group,which is consistent with Lee and his research that APACHE II score is strongly associated with mortality in severely ill patients (*Lee et al., 2017*). However, the *P* value of APACHE II score was more than 0.05 in the multivariable logistic regression analysis for ICU mortality. Therefore, our team did not explain more about APACHE II score, sensitivity and specificity of APACHE II score compare others, nor did we conduct further studies on the prognosis of APACHE II in patients with septic shock.

The causes of poor prognosis of septic shock patients admitted to ICU due to decreased thyroid hormone levels may include immune regulation disorder, coagulation

dysregulation and sepsis related cardiomyopathy (*Soehnlein, 2019*; *Datta & Scalise, 2004*; *Perrotta et al., 2014*; *Montesinos & Pellizas, 2019*). From an evolutionary point of view, the decreased thyroid hormone levels can be interpreted as an attempt to protect the organism by reducing energy consumption and catabolism and may be beneficial in the acute setting (*Ingels, Gunst & Van den Berghe, 2018*). However, several studies have reported that decreased thyroid hormone levels could also cause immune system dysfunction (*Slag et al., 1981*), coagulation system disorder (*Luo, Yu & Li, 2017*), septic cardiac dysfunction (*Iervasi & Nicolini, 2013*), and acute respiratory distress syndrome (*Kim et al., 2018*). Therefore, the question arises whether low thyroid hormone levels should be treated to improve organ function in septic shock. An early experimental study confirmed that thyroid hormone supplementation in septic rats had a beneficial effect and resulted in a lower rate of mortality (*Inan et al., 2003*). Although some clinical studies have also demonstrated that thyroid hormone supplementation may be helpful in critically ill patients (*Kumar et al., 2018*), well-conducted randomized controlled trials are needed to further evaluate the role of thyroid hormone supplementation in septic shock. Furthermore, it is worth noting that no significant difference was found in the duration of mechanical ventilation between the two groups in our study, although the non-survivors had a higher mechanical ventilation rate (48/63 *vs* 61/123, $p = 0.000$). The above phenomenon may be related to the inclusion of non-invasive mechanical ventilation into the category of mechanical ventilation.

However, we could not ignored that there were several limitations in the present study. First, the decrease in serum thyroid hormones levels of septic shock patients is a dynamic process developing over time (*Boelen et al., 2004*). But we only detected the thyroid hormoneone levle in the first 24 h of ICU admission, therefore, we could not observe the changs in thyroid hormone levels of septic shock patients during the ICU treatment. In addition, the result of endocrine-based testing may be related to the timing of sampling. It has been reported that serum thyroid hormone level may require 4 days to reach a nadir after the onset of critical illness (*Woolf et al., 1988*). Finally, dopamine, which can induce iatrogenic hypothyroidism in patients with critical illness (*Van den Berghe, De Zegher & Lauwers, 1994a*; *Van den Berghe, De Zegher & Lauwers, 1994b*; *Schilling et al., 2004*), was not excluded as a confounder in our study.

In conclusion, the data presented in this study pointed out a good performance of thyroid hormone indices in predicting the patients of septic shock, and septic shock patients with higher T3 were associated with good outcome, which can potentially be taken into consideration as new prognostic factors in sepsis shock patients. Of course, the present study has given some insight into the issue and further studies ideally multicentric and prospective with a larger cohort will be needed to corroborate these findings. On all accounts, we should pay more attention to the thyroid hormones of septic shock patients upon admission in ICU.

### Funding
This work was supported by the Nanjing medical science and technology development plan project of China (YKK22115, YKK20114). The funders had no role in study design, data collection and analysis, decision to publish, or preparation of the manuscript.

### Grant Disclosures
The following grant information was disclosed by the authors:
Nanjing medical science and technology development plan project of China: YKK22115, YKK20114.

### Competing Interests
The authors declare there are no competing interests.

### Author Contributions
- Caizhi Sun conceived and designed the experiments, performed the experiments, analyzed the data, prepared figures and/or tables, authored or reviewed drafts of the article, and approved the final draft.
- Lei Bao conceived and designed the experiments, performed the experiments, analyzed the data, authored or reviewed drafts of the article, and approved the final draft.
- Lei Guo performed the experiments, prepared figures and/or tables, authored or reviewed drafts of the article, and approved the final draft.
- Jingjing Wei performed the experiments, authored or reviewed drafts of the article, and approved the final draft.
- Yang Song performed the experiments, authored or reviewed drafts of the article, and approved the final draft.
- Hua Shen performed the experiments, authored or reviewed drafts of the article, and approved the final draft.
- Haidong Qin analyzed the data, authored or reviewed drafts of the article, and approved the final draft.

### Human Ethics
The following information was supplied relating to ethical approvals (*i.e.*, approving body and any reference numbers):

This study protocol was reviewed and approved by the Institutional Review Board of Nanjing Hospital Affiliated to Nanjing Medical University (No. KY20201102-03) and was conducted in accordance with the Declaration of Helsinki.

### Data Availability
The raw data is available in the Supplemental Files.

## Supplemental Information

Supplemental information for this article can be found online at http://dx.doi.org/10.7717/peerj.15335#supplemental-information.

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
