# Peer review of "Prognostic significance of thyroid hormone T3 in patients with septic shock: a retrospective cohort study"

_PeerJ, doi:10.7717/peerj.15335_

## Round 0.1 · original submission · Major Revisions

Please pay special attention to the critical comments of the reviewers, as one of them recommended rejection. Please point out the contribution of the present study, provide the SOFA score and IL-6 data.

Reviewer 1 ·

Basic reporting

No comment

Experimental design

1. Based on your study, it should be “analytical” study, therefore you did analysis by showing up p value
2.Please define the term of “ survivor and non survivor” in methods

Validity of the findings

1.You cannot compare baseline characteristics which is only for descriptive
2.Before doing multivariate, it is better to present the process stages until we found the selected factors to be analyzed.
3.Table 4 should be removed, the ROC ideally presented in graphic.

Additional comments

1.You should explain more about APACHE II score, sensitivity and specificity of APACHE II score compare others.
2.From discussion :
I suggest that the low level of thyroid hormone in this study is due to the septic process, so what we need to do is to treat the sepsis.
3.I think specific population and sample size are not your limitation, on the contrary are the strength of this study.
The more specific the sample, the more homogenous, and the result will be generalizable.

Annotated reviews are not available for download in order to protect the identity of reviewers who chose to remain anonymous.

Reviewer 2 ·

Basic reporting

The results was clear. However, some ambiguous statement in discussion chapter does exist. Author must precisely and carefully interprets the study result, especially when reviewing statistical analysis.
Most references was adequate and contextual.
Background statements that implies the importance and novelty of the study need to be enhanced.
Table and figures was clearly displayed and easy to understand.
The conclusion has already answered the hypothesis.

Experimental design

The research is within the aim and scope of the journal.
Research question was well defined, however, author should enhanced the importance and novelty of the study.
Methods was described with sufficient detail and information to replicate.

Validity of the findings

The impact of the study was assessed. All data have been proved. However, some interpretations related to the results was incorrect, thus may produce improper conclusion.

Additional comments

All of the suggestion was attached in the manuscript.

Annotated reviews are not available for download in order to protect the identity of reviewers who chose to remain anonymous.

·

Basic reporting

- Language is ok
- Text in the figures are extremely small  it is not easy to read (for example Figure 2)
- There are some studies in the literature dealing with the same topic  so the topic is not new: Liu, et al. 2021: “Thyroid hormone disorders: a predictor of mortality in patients with septic shock defined by Sepsis-3?” or Cornu et al. (2020) “Incidence of low-triiodothyronine syndrome in patients with septic shock”

Experimental design

- Retrospective, single-center study
- As mentioned by the authors, it would be nice to have more than one measurement timepoint
- No SOFA score was calculated, when patients were included in the study (although most of the necessary values were included in the table1)

Validity of the findings

- The novelty of the study is questionable (please see above mentioned references)
- Conclusions are correct
- Statistics are correct

Additional comments

- Abstract:
o Please mention in the method section that you enrolled critical ill patients which fulfilling the sepsis criteria, because otherwise the reader is confused what patients are investigated.
o Please explain the abbreviations when using them for the first time, also in the abstract
- Introduction:
o L69: The author described that there is no reliable biomarker for predicting sepsis/septic shock, is this true? I would expect that IL-6 or lactate are useful markers to estimate the development of sepsis/septic shock.
o In the whole manuscript some blank spaces are missing, especially often at the end of the sentences, before the references  please check this.
o L81: As mentioned before, there are studies in the literature describing the same results as the present work; also at an early timepoint.
- Material and Methods:
o L119: It might be interesting for the reader to know the percentages: how many patients had urinary tract infection, how many pulmonary infection…
o Would be interesting to have more than one measurement timepoint, maybe also at admission in the hospital
o L132: we normally using the interleukin 6 on our ICU to detect the early inflammation, because it correlates well with the outcome of the patients. Did you measure the IL-6??
o Did you calculate the SOFA score?
- Results:
o L192: “The mechanical ventilation rate was higher in survivor group”  I would expect that the non-survivors were more often mechanically ventilated
- Discussion:
o l240: “The ICU hospitalization expenses were higher and the length of ICU stay was shorter in the non-survivors”  I would expect that a longer stay at the ICU results in higher expenses in the surviving patients, do you have an explanation for this?
o L272: This is described in the result section completely different  please check again which results are correct.

---

## Round 0.2 · Minor Revisions

The authors have significantly improved the quality of the manuscript. Please provide the minor corrections suggested by reviewer.

Reviewer 2 ·

Basic reporting

The author had revised most of the reviewer's suggestions. The results was clear. Most references was adequate and contextual.
Background statements that implies the importance and novelty of the study need to be enhanced. The author should find the appropriate references/researches that revealed: what the biomarkers had already studied, what the results and the limitations are.
Table and figures was clearly displayed and easy to understand.
The conclusion has already answered the hypothesis.

Experimental design

The research is within the aim and scope of the journal.
Research question was well defined, however, author should enhanced the importance and novelty of the study.
Methods was described with sufficient detail and information to replicate.

Validity of the findings

The impact of the study was assessed. All data have been proved. Conclusions are well mentioned, related to the research question and results.

Additional comments

All of the suggestion was attached in the manuscript.

Annotated reviews are not available for download in order to protect the identity of reviewers who chose to remain anonymous.

·

Basic reporting

language ok; figures improved; As mentioned in the first review round there are some studies in the literature dealing with the same topic

Experimental design

- Retrospective, single-center study
- As mentioned by the authors, it would be nice to have more than one measurement timepoint
- SOFA score and Il-6 values included in the revised manuscript

Validity of the findings

- Conclusions are correct
- Statistics are correct

Additional comments

I think the paper improved a lot. I appreciated the inclusion of the SOFA score and the IL-6 values. Therefore, I would recommend to accept the paper in the present form.

---

## Round 0.3 · accepted · Accept

The authors have improved the manuscript and now it could be accepted for publication.